# Regional Cerebral Oxygen Saturation and Risk of Delirium: A Systematic Review and Meta-Analysis

**DOI:** 10.3390/diseases13120383

**Published:** 2025-11-25

**Authors:** Begoña Rochina-Rodríguez, Francisco Miguel Martínez-Arnau, Pilar Pérez-Ros

**Affiliations:** 1Department of Nursing, Faculty of Nursing and Podiatry, Universitat de València, Menendez Pelayo 19, 46010 Valencia, Spain; begona.rochina@uv.es; 2Department of Physiotherapy, Universitat de València, Gascó Oliag 5, 46010 Valencia, Spain; francisco.m.martinez@uv.es

**Keywords:** oximetry, delirium, spectroscopy, near-infrared, risk, meta-analysis

## Abstract

**Background:** Delirium onset is associated with increased comorbidity and mortality. Identifying reliable delirium biomarkers remains challenging. Regional cerebral oxygen saturation (rSO_2_) offers an objective, easily obtainable measure suitable for hospital monitoring. **Objective:** We aimed to analyse the relationship between regional cerebral oxygen saturation (rSO_2_) values obtained by near-infrared spectroscopy (NIRS) and the subsequent development of delirium. **Methods:** Studies eligible for inclusion in our systematic review evaluated rSO_2_ values obtained by NIRS or a used a similar method to study hospitalised patients aged 18 years or older, some of whom subsequently developed delirium. We searched MEDLINE, Scopus and Web of Science without restrictions to 24 March 2024. Two review authors independently assessed the methodological quality of the included studies using Joanna Briggs Institute Critical Appraisal tools. Using a random-effects model in RevMan v 5.4.0 (Cochrane Collaboration, Oxford, UK), we analysed baseline and minimum rSO_2_ values. Results were presented as means and mean differences (MDs) with their 95% confidence intervals (CIs). We followed PRISMA guidelines and registered our review protocol in PROSPERO (CRD42024523573). **Results (or Findings):** We included 22 studies (20 in the meta-analysis) published between 2009 and 2024 and involving 5757 participants. The delirium group had a lower mean baseline rSO_2_ value (62.47%, 95% CI 58.40 to 66.55) compared with the non-delirium group (64.24%, 95% CI 61.33 to 67.15). Meta-analysis of effect estimates confirmed this result (MD −2.92%, 95% CI −4.38 to −1.47). The MD between the delirium and non-delirium group was larger among patients assessed with the INVOS device and patients who underwent cardiac surgery. Studies that analysed baseline values according to sensor location showed a larger MD in rSO_2_ values obtained via a right-sided sensor. **Conclusions:** Our results show lower baseline and minimum rSO_2_ in hospitalised patients who subsequently developed delirium. The difference varies according to the type of surgery and type of NIRS monitor.

## 1. Introduction

Since the 1990s, near-infrared spectroscopy (NIRS) has been used as a non-invasive method for continuously monitoring the balance between oxygen delivery and consumption in the brain [1]. This technology functions by measuring the absorption of near-infrared (NIR) light (wavelength range 700–950 nm) by molecules called chromophores, which have specific light absorption characteristics in the NIR wavelength [2,3]. NIRS monitors display regional cerebral oxygen saturation (rSO_2_) as a percentage [4].

Currently, NIRS is widely used in surgery and critical care units [5]. Surgical specialties that use the technique to monitor cerebral oxygenation and improve patients’ intraoperative condition include neurological, cardiac, maxillofacial, breast, thoracic, digestive, orthopaedic, and gynaecological surgery [6]. Most studies on the application of NIRS in surgery focus on cardiac procedures such as cardiopulmonary bypass, because they carry a substantial risk of acute intraoperative events that may affect rSO_2_ [7].

In critical care units, NIRS can determine haemodynamic status by measuring the perfusion and oxygenation of the brain, kidney, liver, intestine or muscle, depending on where the sensor is placed. The organ best suited to this technology is the brain, as the tissue is highly dependent on aerobic metabolism, and its function is compromised by both hypoxia and hyperoxia. An alteration in oxygen saturation, the restoration of blood pressure after hypotension or alterations to heart rate can all lead to a severe haemodynamic change, causing encephalic lesions with short- or long-term sequelae [8]. Events that can increase the risk of cognitive dysfunction after surgery or critical illness include increased inflammatory response, hypoperfusion, hypoxia and embolism [9].

Delirium is a neuropsychiatric syndrome frequently seen in acute conditions, particularly in older people. It is characterised by acute onset, a fluctuating course and alterations in consciousness, orientation, memory, thinking, perception and behaviour. The development of delirium is associated with adverse outcomes such as institutionalisation, cognitive decline and death [10].

Delirium incidence in hospitalised people varies according to the setting, ranging from 19% to 64% in medical areas, up to 50% in surgical areas and up to 75% in critical care units. In addition, older people and people with greater comorbidity are at higher risk [11,12]. The underlying aetiology can be classified according to the presence of predisposing factors, which increase an individual’s risk of developing delirium, and participating factors, which trigger the onset of delirium [13].

Poor cerebral perfusion has been implicated in the development of delirium [14]; however, studies examining the association between cerebral perfusion and delirium in high-risk populations (e.g., people undergoing cardiac or abdominal surgery, or ventilated or septic patients in the intensive care unit [ICU]) have shown inconsistent results [15].

Delirium is a severe and very common neuropsychiatric syndrome that increases an individual’s risk of disability and cognitive impairment [16]. Studies have reported a higher incidence of postoperative delirium among people with low rSO_2_ levels before and during surgery [17,18,19]. To our knowledge, only one meta-analysis has quantified the difference in preoperative cerebral SO_2_ values between people who did and did not develop postoperative delirium. It included six studies and found lower values in the delirium group [20]. We aimed to update the pooled evidence on this question because several relevant studies have been published in recent years. Specifically, we aimed to evaluate the relationship between cerebral SO_2_ values obtained by NIRS and the subsequent development of delirium.

## 2. Materials and Methods

This systematic review and meta-analysis included studies comparing rSO_2_ values obtained by NIRS during surgery in people who subsequently developed delirium versus those who did not. Our review protocol is registered in PROSPERO (ID: CRD42024523573).

### 2.1. Inclusion and Exclusion Criteria

Eligible participants were aged 18 years and older. We only included studies that evaluated the association of rSO_2_ values (determined by NIRS or a similar method) with onset of delirium (as assessed by a validated diagnostic method).

Studies in animals, case reports, qualitative studies, letters to the editor, abstracts from conferences, books, systematic reviews and doctoral theses were excluded. In addition, we excluded randomised control trials (RCTs) of delirium prevention interventions, studies that did not quantify rSO_2_ values according to group (delirium and non-delirium) and studies involving patients with previous episodes of emerging delirium or post-traumatic stress disorder and studies without a pure control group due to intervention bias.

### 2.2. Literature Search

Our research question was defined in PECO format:Population: adults aged over 18 yearsExposure: rSO_2_ values obtained using NIRSControl: patients with no deliriumOutcome: delirium

Using this framework, we developed the following search strategy: NIRS OR Near-infrared spectroscopy OR cerebral oximetr* AND delirium OR confusion.

We searched MEDLINE, Scopus and Web of Science from inception to 24 March 2024, using Medical Subject Headings (MeSH) terms in combination with the Boolean operators AND/OR.

### 2.3. Study Selection and Data Collection

All records returned by the literature search were uploaded to a web-based system, which we used to manage the screening process and remove duplicate citations. Two members of the review team (PPR and BRR) independently screened the title and abstract of each unique record to eliminate all records that were clearly irrelevant. The same two review authors then retrieved the full-text articles of the remaining records and assessed them against our inclusion and exclusion criteria. They recorded the study selection process in sufficient detail to create a PRISMA flow chart (Figure 1). To ensure rigour and transparency during data extraction, we employed digital tools that facilitated organisation and reproducibility. We used a customised spreadsheet to systematically record study characteristics, numerical values and statistical estimates, minimising potential errors during synthesis. In parallel, we used the reference management software Zotero 7to organise bibliographic sources. The same two review authors independently extracted the following data from each eligible article:Study details: date of publication, country where study took place, setting, number of patients;Patient characteristics: age, sex, comorbidity;Type of surgery;Details of NIRS: type of NIRS monitor, measurement interval, timing of measurement, sensor location;Baseline NIRS: before induction of anaesthesia with the patient breathing ambient air and 1 min after placement of the measurement sensor;Minimum NIRS: the lowest recorded value during assessment;Maximum NIRS: the highest recorded value during assessment;Outcome: incidence of delirium, method of diagnosis, person who made the diagnosis.

Throughout the study selection and data collection process, disagreements were resolved by consensus on a case-by-case basis. If necessary, a third reviewer was consulted (FMMA).

### 2.4. Study Quality

Two review authors (PPR and BRR) independently assessed the methodological quality of the included studies using the Joanna Briggs Institute (JBI) 10-item checklist for critical appraisal [21]. Each ‘yes’ response scored 1 point, while ‘no’ and ‘unclear’ scored 0 points. A total score of 7 or more points denoted ‘high quality’, 5 to 6 points denoted ‘moderate quality’, and 4 points or fewer denoted ‘low quality’ [22]. The two review authors resolved any disagreements by consensus on a case-by-case basis, consulting a third reviewer (FMMA) if necessary.

### 2.5. Statistical Analysis

After data cleaning in Excel, we conducted statistical analysis using RevMan v5.4.0 (The Cochrane Collaboration, Oxford, UK). Specifically, we pooled the means and standard deviations (SDs) of baseline and minimum rSO_2_ values in the delirium and non-delirium groups according to device and type of surgery. We then combined the mean differences (MDs) between the groups according to NIRS monitoring device, type of surgery, and sensor placement. Where studies did not report MDs or percentages, we calculated them if there were sufficient data available. To quantify heterogeneity, we used the I^2^ statistic.

## 3. Results

### 3.1. Characteristics of Studies

The search returned 589 records, of which 288 were duplicates, and 227 were excluded during the title and abstract screen. Of the remaining 74 records, 65 were available as full-text articles. During the full-text review, 22 articles were deemed eligible for inclusion. Of the articles excluded at this stage, six had no non-delirium group [5,15,23,24,25,26], three were reviews [27,28,29], 13 were RCTs [30,31,32,33,34,35,36,37,38,39,40,41,42], seven included no assessment of delirium [43,44,45,46,47,48,49], six provided no rSO_2_ data [11,50,51,52,53,54], and four were other ineligible document types [55,56,57,58].

After the selection process, we included 22 studies in the systematic review and 20 in the meta-analysis. The included studies were published between 2009 and 2024 and involved a total of 5757 participants. The mean participant age ranged from 52.62 years [59] to 83 years [60], and there were more men than women in most studies. Table 1 provides further details.

One study was set in the ICU [19], and all other studies were conducted in a surgical setting. Thirteen studies (59%) involved cardiac and vascular surgeries [17,59,61,62,63,64,65,66,67,68,69,70,71], four (18%) involved abdominal surgery [72,73,74,75], one (4.5%) involved thoracic surgery [76], two (9%) involved orthopaedic surgery [60,77], and one (4.5%) included other surgical procedures (general surgery) [18].

To detect and diagnose delirium, 15 studies (68%) used the Confusion Assessment Methods (CAM) scale [18,59,60,61,70,71,72,75,76] and/or its variants: the CAM-ICU [17,19,61,64,65,67,70,71,76,77,78], Intensive Care Delirium Screening Checklist (ICDSC) [77], and the chart-based method [69]. Three studies used Diagnostic and Statistical Manual of Mental Disorders (DSM) criteria [24,73,74]. In the remaining three studies, the diagnosis was based on symptoms such as confusion, agitation or change in mental state [62,66,69]. The professionals responsible for diagnosing delirium were physicians and neuropsychiatrists [24,67,74], nurses [59,64,66] and trained research personnel [17,18,19,60,61,65,71,72,76]. This information was missing from seven studies [62,69,70,73,75,77,78].

The proportion of participants who experienced delirium ranged from 7.1% [75] to 78.4% [19]. To measure rSO_2_, 15 studies (68%) used the INVOS monitor [17,18,60,61,62,64,66,67,69,70,71,72,73,75,77], four used FORESIGHT Elite [19,63,65,76], one used INT-100 (Hefei ENO Electronics) [74], and one used C2030C (CAS Medical Systems) [59]. One study did not specify monitor type [68].

Fourteen studies (63.6%) monitored rSO_2_ only during surgery [59,60,61,64,66,67,69,71,72,73,74,75,76,77], four continued monitoring in the hours following surgery [62,63,65,70], and three also monitored rSO_2_ the previous day [17,18,68]. Finally, in the study conducted in the ICU, monitoring took place over 24 h [19]. Monitoring began from the first minute after sensor placement, this first value being considered the baseline. Seven studies used continuous monitoring [17,60,62,69,71,75,77], and 10 studies recorded values at 2 s, 2 min or 5 min intervals [19,59,61,65,66,67,70,72,73,76]. One study measured rSO_2_ over 90 min each day of ICU admission [63], while two studies performed the measurement at pre-established time intervals [64,74]. Two studies provided no information on the time interval of monitoring [18,68].

**Table 1 diseases-13-00383-t001:** Characteristics of included studies.

Study ID	Country	Surgery	No. of Patients/Age (Years)/% Male		Morbidity	% Delirium	Diagnostic Tool/ Professional	NIRS Monitor	Measurement Interval	Moment	Forehead Sensor Placement
			Delirium	No Delirium							
Ahn 2021 [61]	Korea	Cardiac	n = 230 M 62.6 (SD 13.7) 51.7%	n = 460 M 62 (SD 14.6) 51.3%	NA	33.3%	CAM and CAM-ICU/Trained or experienced personnel	INVOS	5 min intervals after 1 min post sensor	During surgery	Bilateral
Bennett 2021 [62]	Saudi Arabia	Cardiac	n = 14	n = 152	NA	8.4%	Documented in medical notes and prescribed haloperidol/NA	INVOS 5100	Continuous	Before induction to ICU	Bilateral
			M 8.4 ^a^ 73% ^a^								
Chan 2019 [63]	Australia	Cardiac	n = 24 Mdn 69 (IQR 64–77) 79.2%	n = 84 Mdn 66 (IQR 57–71) 71.4%	EuroSCORE, Mdn (IQR): 2.8 (1.9–4.2) vs. 2.2 (1.4–3.1) APACHE 3, Mdn (IQR): 58 (49–67) vs. 47 (39–53) *** SAPS II, Mdn (IQR): 51 (45–54) vs. 46 (42–51)	22.2%	CAM-ICU/NA	FORESIGHT Elite	90 min each day	ICU	Bilateral
Chen 2024 [64]	China	Cardiac	n = 47 M 70 (SD 5) 51.1%	n = 85 M 70 (SD 4) 50.5%	NA	35.6%	CAM-ICU/CCU nurse specialist	INVOS 5100C	4 intraoperative time points ^b^	During surgery	Bilateral
Clemmesen 2018 [60]	Denmark	Trauma	n = 10	n = 30	ASA I: n = 3 (7.5%) ASA II: n = 30 (75.0%) ASA III: n = 5 (12.5%) ASA IV: n = 2 (5%) ^a^	25%	MDAS and CAM/Trained research personnel	INVOS 5100	Continuous	During surgery	Right
			Mdn 83 (IQR 78–89) ^a^ 10% ^a^								
Cui 2021 [76]	China	Thoracic	n = 35	n = 140	ASA I: n = 1 (0.6%) ASA II: n = 136 (77.7%) ASA III: n = 38 (21.7%) ^a^	20%	CAM and CAM-ICU/Trained research personnel	FORESIGHT ELITE	2 sec intervals	During surgery	Bilateral
			M 64.5 (SD 6.4) ^a^ 52% ^a^								
Eertmans 2020 [65]	Belgium	Cardiac	n = 29 Mdn 79 (IQR 75–83) 19%	n = 67 Mdn 75 (IQR 73–79) 45%	EuroSCORE II, Mdn (IQR): 2.61 (1.75–4.68) vs. 1.86 (1.02–3.37) *	30%	CAM ICU/Trained research personnel	FORESIGHT Elite	2 sec intervals	After induction to 72 h after surgery	Bilateral
Fischer 2022 [72]	Germany	Abdominal	n = 13	n = 80	NA	16.3%	3D-CAM/Research personnel	INVOS 5100C	1 min intervals	During surgery	NA
			M 66.31 (SD 10.55) ^a^ 58.7% ^a^								
Hori 2014 [66]	Japan	Cardiac	n = 45 M 69.6 (SD 9.9) 82%	n = 446 M 65.8 (SD 11. 4) 72.2%	NA	9.2%	Presence of confusion, agitation or change in mental status/Nurses	INVOS	10-sec intervals	Before induction of anaesthesia	Bilateral
Lim 2020 [67]	Korea	Cardiac	n = 105 M 71.9 (SD 8.2) 70.5%	n = 710 M 65.2 (SD 9.6) 78.3%	ASA I: n = 1 (1%) vs. n = 19 (2.7%) ASA II: n = 25 (23.8%) vs. n = 200 (28.2%) ASA III: n = 75 (71.4%) vs. n = 478 (67.3%) ASA IV: n = 4 (3.8%) vs. n = 13 (1.8%)	14.8%	CAM-ICU/Neuropsychiatrist	INVOS	5 min intervals	During surgery	Bilateral
Mailhot 2019 [68]	Canada	Cardiac	n = 173 Mdn 74 (IQR 68–74) 73.4%	n = 173 Mdn 69 (IQR 61–75) 76.3%	EuroSCORE, Mdn (IQR): 3.49 (1.96–5.38) vs. 2.20 (0.84–3.43) * NYHA ≥ 3: n = 8 (4.6%) vs. n = 15 (8.7%)	50%	DSM-V/Physician	NA	NA	Pre-operative	NA
Momeni 2019 [69]	Belgium	Cardiac	n = 303 Mdn 75 (IQR 64–80)	n = 1201 Mdn 67 (IQR 57–75)	EuroSCORE II, Mdn (IQR): 2.64 (1.42–4.92) vs. 2.25 (1.19–3.86) **	20.14%	Validated chart review method searching in the medical record/NA	INVOS 5100	Continuous	During surgery	Bilateral
			71% ^a^								
Morimoto 2009 [73]	Japan	Abdominal	n = 5 M 76 (SD 4) 75%	n = 15 M 68 (SD 3) 66%	NA	25%	DSM-IV/NA	INVOS 3100	1 min intervals	During surgery	Left
Nakano 2021 [70]	Japan	Cardiac and ICU	n = 22	n = 112	EuroSCORE, Mdn (IQR): 3.45 (1.72–6.09) ^a^	16.4%	3D-CAM and CAM-ICU daily/NA	INVOS	10-sec intervals	During surgery and the day after	Bilateral
			Mdn 65 (IQR 58–71) ^a^ 74.6% ^a^								
Schoen 2011 [17]	Germany	Cardiac	n = 62 M 73.1 (SD 6.7)/54.8%	n = 169 M 64.9 (IQR 13.3)/68%	EuroSCORE, M (SD): 7.9 (3.7) vs. 5.9 (3.5)	26.8%	CAM-ICU/Trained research personnel	INVOS	Continuous	During surgery	Bilateral
Soh 2016 [77]	Korea	Trauma	n = 9 Mdn 73 (IQR 70–77) 44%	n = 100 Mdn 75 (IQR 72–77) 52%	NA	8%	ICDSC and CAM-ICU/NA	INVOS	Continuous	During surgery	Bilateral
Soh 2020 [71]	Korea	Cardiac	n = 16 M 71 (SD 5) 69%	n = 97 M 70 (SD 6) 62%	EuroSCORE, Mdn (IQR): 7 (5–9) vs. 6 (4–8)	14.16%	CAM-ICU and CAM/Trained research personnel	INVOS	Continuous	During surgery	Bilateral
Song 2022 [74]	China	Major abdominal	n = 16 Mdn 75 (IQR 72–80.5) 56.3%	n = 85 Mdn 72 (IQR 65–77) 70.6%	CIRS, Mdn (IQR): 11.5 (7.5–14) vs. 7 (7–10) **	15.8%	DSM-IV/Physician	INT-100, Hefei ENO Electronics	6 intraoperative time points ^c^	During surgery	Bilateral
Susano 2021 [18]	Portugal	Elective surgical procedures	n = 53 Mdn 76 (IQR 71–80) 60%	n = 185 Mdn 72 (IQR 68–77) 68%	ASA ≥ III: n = 35 (66%) vs. n =67 (36%)	22.2%	CAM/Trained research personnel	INVOS 5100C	1 min post sensor	Pre-operative	Bilateral
Tobar 2018 [75]	Chile	Abdominal	n = 2	n = 26	ASA I: 35.7% ASA II: 64.3%	7.1%	CAM/NA	INVOS 5100	Continuous	During surgery	Bilateral
			M 73 (SD 7) ^a^ 39.3% ^a^								
Wang 2019 [59]	China	Cardiac	n = 14 Mdn 54.1 (IQR 46.2–62) 17.9%	n = 25 Mdn 52.6 (IQR 47.7–57.5) 46.2%	ASA, M (95% CI): 2.9 (2.1–3.1) vs. 2.8 (2.7–3.5)	35.8%	CAM/Trained nurse	C2030C, CAS Medical Systems	1 min intervals	During surgery	Bilateral
Wood 2017 [19]	Canada	General medical/surgical and trauma ICU	n = 19 Mdn 71 (IQR 67–76) 79%	n = 69 Mdn 68 (IQR 54–77) 59%	APACHE, Mdn (IQR): 21 (15–27) vs. 20 (18–23)	78.4%	CAM-ICU/Trained researchers	FORESIGHT Elite	2 sec intervals	24 h	Bilateral

^a^: all sample data; ^b^: start of anaesthesia (T1), start of cardiopulmonary bypass (T2), start of rewarming (T3), and end of cardiopulmonary bypass (T4); ^c^ Before induction of anaesthesia (T0), 5 min after induction of anaesthesia (T1), 5 min after PaCO_2_ (partial pressure of arterial carbon dioxide) reached 25–30 mmHg (T2), 5 min after PaCO_2_ reached 45–50 mmHg (T3), at the end of surgery (T4), 5 min after PaCO_2_ reached 25–30 mmHg again (T5), and 5 min after PaCO_2_ reached 45–50 mmHg again (T6). Abbreviations: APACHE: Acute Physiology and Chronic Health Evaluation; ASA: American Society of Anesthesiologists; CAM: Confusion Assessment Method; CCU: critical care unit; CI: confidence interval; CIRS, cumulative illness rating scale; DSM-IV: Diagnostic and Statistical Manual of Mental Disorders, Fourth Edition; DSM-V: Diagnostic and Statistical Manual of Mental Disorders, Fifth Edition; EuroSCORE: European System for Cardiac Operative Risk Evaluation; ICDSC: Intensive Care Delirium Screening Checklist; ICU: intensive care unit; IQR: interquartile range; M: mean; Mdn: median; MDAS: Memorial Delirium Assessment Scale; NA: not available; rSO_2_: regional oxygen saturation; SAPS II: Simplified Acute Physiology Score II; 3D-CAM; 3 min diagnostic interview for confusion assessment method; * *p* < 0.05; ** *p* < 0.01; *** *p* < 0.001.

Sensor placement for rSO_2_ measurement was bilateral in 17 studies (77.2%) [17,18,19,59,61,62,63,64,65,66,67,69,70,71,74,75,76,77], right-sided in one study [60] and left-sided in one study [73]. Two studies did not specify where the sensors were placed [68,72].

### 3.2. Oximetry Values

Fifteen studies reported baseline rSO_2_ values and were included in our primary analysis (Table 2).

We calculated pooled baseline values for both groups, finding a lower value for the delirium group (62.47%, 95% CI 58.40 to 66.55; Tau^2^ = 57.98; Chi^2^ = 832.82, df = 14, *p* < 0.001; I^2^ = 98%; 15 studies, 842 participants, Appendix A) compared with the non-delirium group (64.24%, 95% CI 61.33 to 67.15; Tau^2^ = 30.42; Chi^2^ = 886.98, df = 14, *p* < 0.001; I^2^ = 98%; 15 studies, 2598 participants; Appendix A). The lowest values were reported in studies that used the INVOS monitor (delirium: 59.91%, 95% CI 53.96 to 65.87; non-delirium: 61.89%; 95% CI 59.02 to 69.76; Appendix A).

Meta-analysis of effect estimates confirmed a lower baseline rSO_2_ in the delirium group versus the non-delirium group (MD −2.92%, 95% CI −4.38 to −1.47; Tau^2^ = 4.58; Chi^2^ = 47.02, df = 14, *p* < 0.0001; I^2^ = 70%; 15 studies, 3430 participants; Figure 2). We found a greater difference when we included only studies that had used the INVOS monitor (MD −3.70%, 95% CI −5.62 to −1.79; Tau^2^ = 5.58; Chi^2^ = 24.62, df = 9, *p* < 0.0001; I^2^ = 74%; 10 studies, 2602 participants; Figure 2a) and only studies that had involved cardiac surgeries (MD −3.37%, 95% CI −5.28 to −1.47; Tau^2^ = 5.35; Chi^2^ = 47.02, df = 14, *p* < 0.0001; I^2^ = 79%; 8 studies, 2601 participants; Figure 2b).

We also analysed MD according to the hemisphere of sensor location in five studies that had used the INVOS device. The delirium group had lower rSO_2_ values as measured both in the right hemisphere (MD −4.61%, 95% CI −7.47 to −1.74; Tau^2^ = 5.31; Chi^2^ = 12.19, df = 3, *p* < 0.007; I^2^ = 75%; 4 studies, 2725 participants; Appendix A) and the left hemisphere (MD −3.80%, 95% CI −6.40 to −1.21; Tau^2^ = 5.01; Chi^2^ = 15.31, df = 3, *p* < 0.002; I^2^ = 80%; 4 studies, 2710 participants; Appendix A), through the difference was greater in the right hemisphere measurements. The subgroup analysis by surgical specialty showed lower rSO_2_ values (as measured in both the right and left hemispheres) in the delirium group in studies that involved cardiac, abdominal and orthopaedic surgery, through the difference was greater in the right hemisphere (Appendix A).

When we analysed the minimum reported rSO_2_ values, we found a lower pooled mean minimum in the delirium group (50.67%, 95% CI 46.82 to 54.53; Tau^2^ = 24.92; Chi^2^ = 4153.78, df = 7, *p* < 0.00001; I^2^ = 95%; 8 studies, 630 participants, Appendix A) than in the non-delirium group (54.84%, 95% CI 51.91 to 57.78; Tau^2^ = 15.15; Chi^2^ = 273.44, df = 7, *p* < 0.00001; I^2^ = 97%; 8 studies, 1806 participants). The lowest values were reported in studies that used the INVOS device, for both the delirium group (47.25%, 95% CI 45.48 to 49.02; Appendix A) and the non-delirium group (53.19%, 95% CI 49.92% to 56.46%; Appendix A). When we analysed the minimum values according to type of surgery, we found lower values from studies that involved cardiac surgery, for both the delirium group (50.32%, 95% CI 46.19 to 54.45; Appendix A) and the non-delirium group (54.36%, 95% CI 51.05 to 57.67; Appendix A).

Meta-analysis of effect estimates confirmed the lower minimum rSO_2_ values in the delirium group compared with the non-delirium group (MD −4.22%, 95% CI −7.08 to −1.36; Tau^2^ = 10.57; Chi^2^ = 40.19, df = 7, *p* < 0.00001; I^2^ = 83%; 8 studies, 2440 participants; Appendix A). We found a greater difference in the studies that used the INVOS device (MD −5.33%, 95% CI −8.10 to −2.57; Tau^2^ = 7.06; Chi^2^ = 23.47, df = 5, *p* < 0.0003; I^2^ = 79%; 6 studies, 1998 participants; Appendix A) and the studies that involved cardiac surgeries (MD −4.27, 95% CI −7.32 to −1.23; Tau^2^ = 11.24; Chi^2^ = 39.71, df = 5, *p* < 0.00001; I^2^ = 87%; 6 studies, 2291 participants; Appendix A).

Finally, we conducted the same analyses according to the type of measurement (continuous or intermittent). There was a higher MD in rSO_2_ values with continuous assessment (MD −4.15, 95% CI −6.95 to −1.35; Tau^2^ = 3.37; Chi^2^ = 6.37, df = 4, *p* < 0.00001; I^2^ = 37%; 5 studies, 647 participants) compared to intermittent assessment (MD −2.54, 95% CI −4.51 to −0.57; Tau^2^ = 6.11; Chi^2^ = 36.36, df = 8, *p* < 0.00001; I^2^ = 78%; 9 studies, 2437 participants), also reflecting the heterogeneity of the types of observation measures (Appendix A). The results were similar in the remaining analyses for baseline and minimum values (Appendix A).

**Figure 2 diseases-13-00383-f002:**
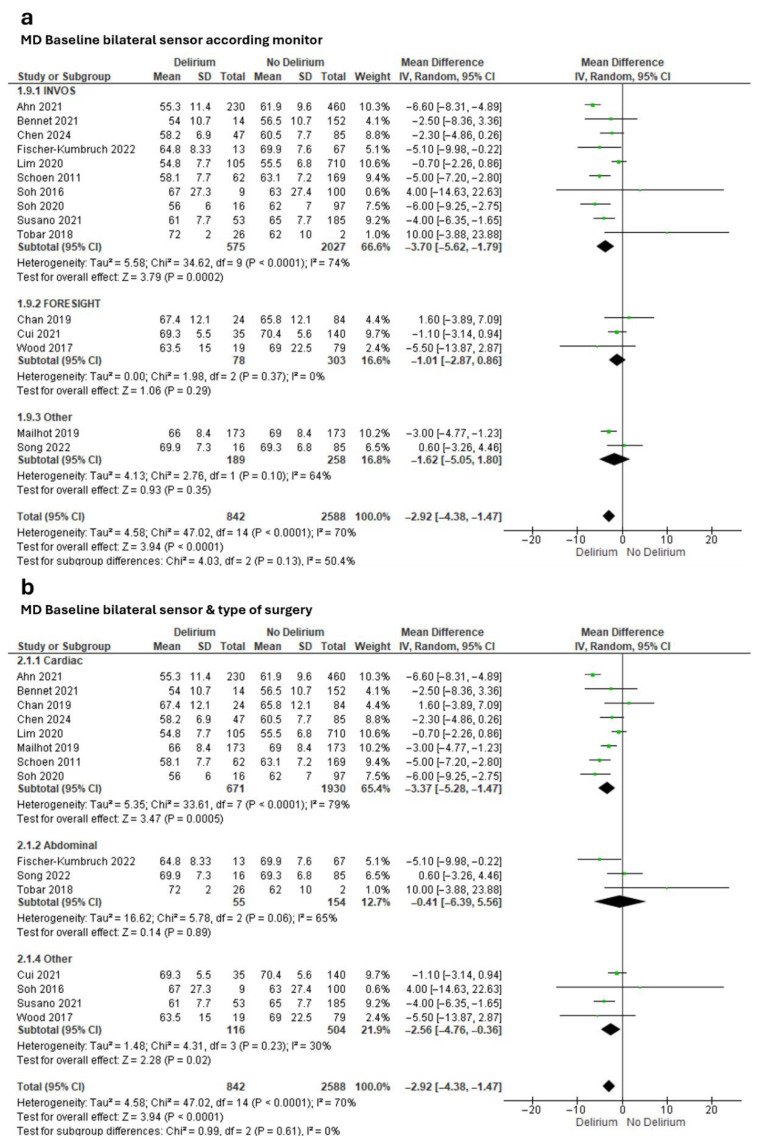
Meta-analysis of mean baseline cerebral oxygen saturation values obtained through bilateral sensors in people with and without subsequent delirium according to (**a**) monitor and (**b**) type of surgery [17,18,19,61,62,63,64,67,68,71,72,74,75,76,77].

### 3.3. Methodological Quality of Included Studies

Overall, we rated the methodological quality of the included studies as moderate to high using the JBI Critical Appraisal Checklist for Cohort Studies. Some studies failed to identify confounding factors, to deal with confounding factors and/or to address incomplete follow-up. Table 3 presents the ratings for all studies.

When we created funnel plots for our subgroup analyses, we found a few studies were outside the funnel, possibly due to small sample sizes. Most studies were distributed at the top of the funnel, which indicates greater precision (Appendix A).

## 4. Discussion

The pathophysiology of delirium is complex and multifactorial, involving alterations in neurotransmission, neuroinflammation, oxidative stress and blood–brain barrier dysfunction [15]. In this context, cerebral hypoxia emerges as a key precipitating factor, particularly in patients with systemic or cerebral oxygenation compromise. Cerebral oxygenation depends on a delicate balance between oxygen supply and demand, primarily regulated by cerebral blood flow and cerebral perfusion pressure [79]. In conditions such as acute brain injury or prolonged mechanical ventilation, silent cerebral hypoxia may occur even in the absence of systemic hypoxemia or elevated intracranial pressure. This hypoxia can trigger a cascade of pathophysiological events, including mitochondrial dysfunction, free radical accumulation, pro-inflammatory cytokine release, and neurotransmitter imbalances, particularly in the cholinergic and dopaminergic systems [80]. Recent studies have demonstrated that both acute and chronic hypoxia can lead to significant cognitive impairment, affecting attention, memory, and executive function—all domains commonly disrupted in delirium. Moreover, prolonged hypoxia has been associated with hippocampal and cortical atrophy, as well as structural changes such as amyloid plaque deposition and neurofibrillary tangles, suggesting a potential link between delirium and neurodegenerative processes [80].

Some systematic reviews with meta-analyses have investigated the risk of delirium after NIRS monitoring versus no monitoring [81,82,83,84,85]. They found that NIRS monitoring reduced postoperative cognitive dysfunction but were unable to draw solid conclusions regarding the risk of postoperative delirium, stroke, cardiac dysfunction, length of stay and mortality. In contrast, some recent studies have shown that people with low cerebral SO_2_ levels before and during surgery have a higher incidence of postoperative delirium [15,17,18,19,55]. We aimed to update the pooled evidence on this question because several relevant studies have been published in recent years. Specifically, we aimed to evaluate the relationship between cerebral SO_2_ values obtained by NIRS and the subsequent development of delirium. We found lower baseline and minimum rSO_2_ values in the delirium group.

Almost all studies included in our review were conducted in a surgical setting, most frequently cardiac surgery. NIRS is widely used in cardiac surgery owing to the high risk of cerebral hypoxia and haemodynamic instability, which increase the vulnerability of the brain and other vital organs [4]. In the study populations, older age and male sex predominated, which is consistent with the higher prevalence of cardiac pathology in older men [86]. Most studies reported comorbidity indices, but only four found significant differences, with higher comorbidity in the delirium group. This suggests comorbidity could act as a risk factor for delirium independently of oximetry values [87]. To enable better comparisons, future studies should present separate comorbidity levels for each group.

The percentage of participants who developed delirium across the included studies ranged from 7.1% to 78.4%. According to the literature, 24% of older people develop postoperative delirium [88], although the risk may vary depending on the type of surgery [89]. The presence of predisposing risk factors prior to assessment may also influence the incidence of delirium. Although some studies propose excluding patients with predisposing risk factors to determine the true utility of cerebral oximetry [7], we argue that including these patients and stratifying the analysis by risk factor may be more useful. This approach acknowledges the multifactorial nature of delirium and the clinical reality that predisposing factors elevate risk. Surgery itself is a precipitating factor, as is the ICU setting [90]. Most included studies did not indicate the specific time point of delirium diagnosis during admission. The main method of screening for delirium was the CAM scale, in line with most similar studies [7,90], as it is easy to administer and has excellent diagnostic accuracy [91].

Cerebral SO_2_ values are generally assessed through two sensors placed on the forehead. The bilateral baseline values are the most crucial results from cerebral oximetry and should be acquired prior to the administration of oxygen or any drugs that may affect these values. Normal rSO_2_ levels are around 60% to 80%, as cerebral arterial blood has an oxygen saturation of 98% to 100%, venous blood has an oxygen saturation of around 60%, and the ratio of arterial to venous blood is between 70:30 and 75:25. It is also important to measure the percentage variation from baseline throughout the surgical procedure, as values that fall outside of this range may be considered normal [4]. In our analysis, the pooled baseline rSO_2_ values were 62.47% in the delirium group and 64.24% in the non-delirium group. The baseline value reported by Chan and colleagues in their systematic review was 66.4% (95% CI 65.0 to 67.7); based on this result, the authors proposed a normal baseline preoperative range of 51.0% to 81.8% [92].

Cerebral oximetry values are related to age as well as the presence of cardiovascular disease, cerebral atrophy, smoking and other factors [93]. In our subgroup analysis by type of surgery, the lowest pooled baseline rSO_2_ value was observed in cardiac surgery patients (58.89% in delirium vs. 61.76% in non-delirium). This finding is in line with the literature, which associates cardiovascular disease with decreased cerebral oxygenation [93]. Lower rSO_2_ values are considered an indicator of cognitive frailty or poor brain reserve, which predisposes individuals to a greater risk of later cognitive impairment [94]. In addition, our subgroup analysis by type of NIRS monitor showed lower baseline rSO_2_ in people evaluated with the INVOS monitor in both groups. (59.91% in delirium vs. 61.89% in non-delirium). Most studies used the INVOS monitor because it has been on the market for more than 20 years. Differences with other monitors include the duration of sensor placement and type of light [93]. Therefore, analyses should be stratified according to manufacturer to avoid biases in the assessment method [93,95].

The forest plots reveal considerable heterogeneity among the included studies, as indicated by I^2^ values of 99% and 98% and highly significant Chi^2^ tests (*p* < 0.00001). This suggests that the variability in effect sizes is not due to chance but reflects true differences across studies. Potential sources of heterogeneity include differences in patient populations (age, comorbidities, baseline cognitive status), variability in monitoring devices (INVOS vs. FORE-SIGHT) with distinct calibration and sensitivity profiles, and differences in clinical contexts such as surgical type and perioperative management. Additionally, inconsistencies in baseline measurement timing and protocols may have contributed to the wide confidence intervals observed. Although subgroup analysis by device type partially explains this variability (*p* = 0.05), residual heterogeneity remains substantial, indicating that unmeasured confounders or methodological differences persist.

Meta-analysis of 15 studies showed an MD between the groups of nearly three percentage points in baseline rSO_2_. This differed according to monitor type, with larger MDs observed in studies that used the INVOS monitor and in studies that involved cardiac surgery. Clinically, a difference of nearly 3% in NIRS values may seem modest, but in the context of cerebral autoregulation, this variation may reflect a critical decrease in cerebral perfusion [92]. Because the brain has high metabolic demand and low tolerance to hypoxia, even small sustained reductions in oxygenation could trigger neuronal dysfunction and contribute to the development of delirium [94]. Furthermore, as ours is the first meta-analysis to report this difference, it serves as a starting point for quantifying significantly lower values in patients who develop delirium, encouraging further prospective studies to report more complete data in a larger sample.

We also observed a greater difference in continuous assessment compared to intermittent assessment, with the latter showing greater heterogeneity. This may be due to the different protocols used, capturing intervals of 2 s, 5 min, or at different times during the intervention. Continuous assessment is more sensitive, detecting small changes with greater reliability, which could explain the values obtained in the meta-analysis [96].

When we analysed studies that had reported baseline values by sensor location (right or left forehead), we found a slightly larger MD in rSO_2_ obtained via the right forehead sensor. Because the right hemisphere is responsible for attention, lesions in this side of the brain are more likely to lead to delirium [97]. Although rSO_2_ under normal conditions is mainly symmetrical, differences of around 10% are considered physiological [98]. Furthermore, studies indicate no differences in rSO_2_ obtained through one sensor (regardless of hemisphere) or two, unless there is in situ pathology [92,99].

Several included studies reported the minimum rSO_2_ value obtained during assessment. The pooled mean minimum value was lower in patients who developed delirium (50.67%) versus those who did not (54.84%), with an MD of −4.22% (95% CI −7.08 to −1.36). While we performed subgroup analyses of minimum values according to NIRS monitor and type of surgery, substantial between-study heterogeneity limited meaningful comparison of the results. It is more complex to detect minimum values through intermittent assessment compared with continuous assessment. In addition, beyond the minimum value, it is important to assess the reduction from baseline, as the current literature defines cerebral desaturation through two criteria: absolute values below 50% (range 40% to 60%) or relative reductions exceeding 20% (range 10% to 30%) with respect to the baseline [94]. We were unable to perform a meta-analysis according to desaturation cut-off value due to the lack of unified criteria to compare them. Some studies provided an absolute value and others a percentage of desaturation. Some even used different cut-off values for the right and left sensor. Although surgery, especially cardiac surgery, is a hemodynamic risk factor, patients can maintain near-baseline hemodynamic conditions if there are no stressors during anaesthesia [89,100]. However, in our review, lack of data precluded analysis of the surgical protocol, medication administered, management of fraction of inspired oxygen (FiO_2_) and blood pressure or duration of surgery. Future RCTs and other longitudinal studies should take these factors into account in their analyses [81].

Despite methodological challenges in cross-study comparisons, we found lower baseline and minimum rSO_2_ values in people who developed delirium. These findings provide a basis for future studies to quantitatively characterise cerebral desaturation during surgery and at delirium onset.

## 5. Limitations

The first limitation of this review is that our search was restricted to the predefined terms and selected databases, which means we may have missed some relevant studies published in other sources or using different terminology. We acknowledge that the included studies may not constitute an exhaustive representation of all relevant research. The different tools of detecting delirium could also represent a bias in identification. Second, it was not possible to include all the articles in the subgroup analyses because some provided insufficient data. We contacted the study authors to request these data but received no reply. Third, there was a lack of information on some confounding factors, such as comorbidities (e.g., diabetes and hypertension, which can lead to narrowed cerebral arteries through glycosylation and vasculitis), predisposing and precipitating risk factors, and time from NIRS assessment to delirium onset. It was not possible to perform a systematic check for rSO_2_ symmetry or apply corrections for individual anatomical factors due to the design and scope of the study. Likewise, we could not rule out the potential influence of local hemodynamic or technical artefacts (e.g., oedema, internal carotid artery atherosclerosis or variations in sensor pressure on the skin), which may have affected the interpretation of the observed asymmetry. Fourth, there was substantial between-study heterogeneity in the method of cerebral oximetry assessment, stemming from variation in the monitor used, the data collection protocol (continuous or intermittent), and other factors. Finally, the funnel plot assessment indicated possible publication bias.

We recommend that future studies provide detailed participant characteristics for both groups, including demographic data (age and sex), a validated comorbidity index, predisposing risk factors (e.g., diabetes, dementia, stroke history) and medication profile. It is important to describe how the oximetry data is collected (continuous or intermittent), specify the NIRS monitor used and provide mean values for both groups at baseline with SDs. In the surgical setting, researchers should report the minimum value observed, quantifying percentage desaturation or decline from baseline and indicating the time of the observation together with total surgery time. Complete reporting should also include the postoperative care setting, delirium assessment methodology (instrument, assessment frequency, duration of follow-up) and timing of delirium diagnosis. Finally, reporting complications and duration of admission enables adjustment for confounding factors.

## 6. Conclusions

Baseline and minimum regional cerebral oxygen saturation (rSO_2_) values obtained via near-infrared spectroscopy (NIRS) were lower among people who subsequently developed delirium compared with those who did not. We found that rSO_2_ values varied according to the type of surgery and type of NIRS monitor, with the lowest values among people who underwent cardiac surgeries and those assessed with INVOS monitors. The studies that analysed baseline values according to sensor location showed a greater mean difference in rSO_2_ obtained with the right sensor compared with the left. This is a starting point for studies aiming to investigate the exact nature of cerebral oxygen decline in surgical patients and its relationship to delirium onset. Given that rSO_2_ is an objective and easily obtainable measurement, its routine use in hospital settings could improve early detection of delirium risk and optimise clinical decision-making. Studies in other community institutional settings are also warranted.

## Figures and Tables

**Figure 1 diseases-13-00383-f001:**
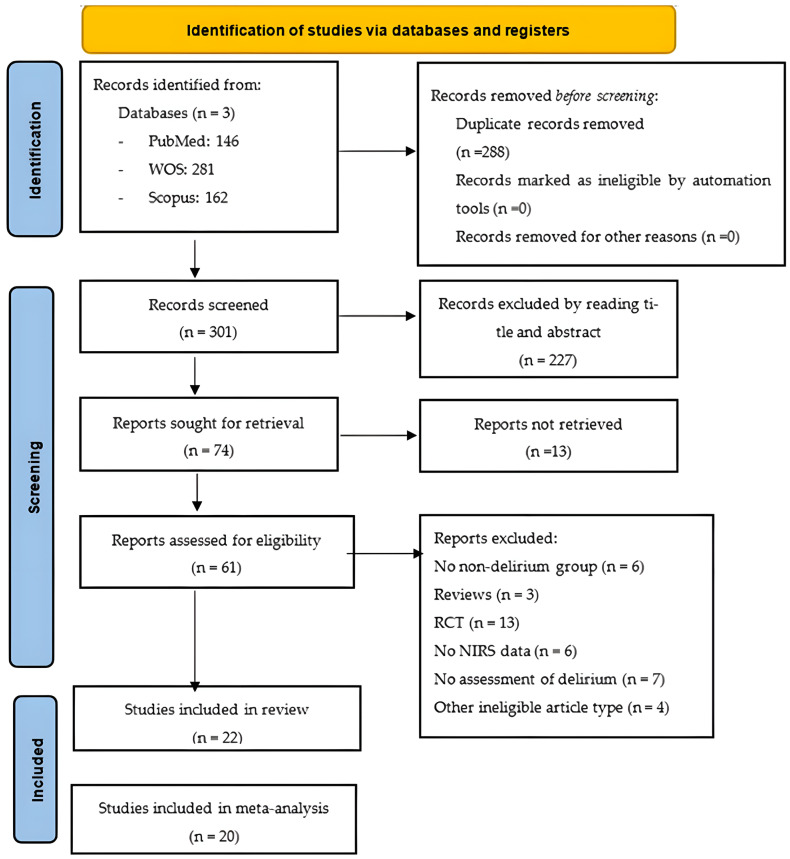
PRISMA flow chart. Abbreviations: NIRS: near-infrared spectroscopy; RCT, randomised controlled trial.

**Table 2 diseases-13-00383-t002:** Baseline, minimum and maximum cerebral oxygen saturation values in people with and without subsequent delirium.

	Baseline rSO_2_ (%) Mean (SD)/Median (IQR)		Minimum rSO_2_ (%) Mean (SD)/Median (IQR)		Maximum rSO_2_ (%) Mean (SD)/Median (IQR)	
Study ID	Delirium	No Delirium	Delirium	No Delirium	Delirium	No Delirium
bilateral						
Ahn 2021 [61]	55.3 (11.4)	61.9 (9.6)	48.3 (10.5)	52.2 (8.3)	73.2 (9.2)	72.7 (8.3)
Bennett 2021 [62]	54 (4) ^a^	56.5 (2) ^a^	—	—	—	—
Chan 2019 [63]	67.4 (62.2–70.6)	65.8 (62.1–72)	—	—	—	—
Chen 2024 [64]	58.2 (6.9)	60.5 (7.7)	—	—	—	—
Clemmesen 2018 [60]	60.5 (58–75)	68.5 (61–73)	51.5 (46–63)	58.5 (50–65)	—	—
Cui 2021 [76]	69.3 (5.5)	70.4 (5.6)	—	—	—	—
Eertmans 2020 [65]	68 (65–69)	68 (66–70)	60 (55–62)	59 (55–62)	—	—
Fischer 2022 [72]	64.8 (8.3)	69.9 (7.6)	—	—	—	—
Lim 2020 [67]	54.8 (7.74)	55.5 (6.8)	46.7 (8.33)	55.5 (6.8)	—	—
Mailhot 2019 [68]	66 (60–70)	69 (64–73)	58 (52–63)	63 (58–67)	—	—
Morimoto 2009 [73]	59.5 (4.5) ^a^	66 (8) ^a^	—	—	—	—
Schoen 2011 [17]	58.1 (7.7)	63.1 (7.2)	48.6 (9.3)	55.1 (8.6)	—	—
Soh 2016 [77]	67 (54–70)	63 (59–68)	55 (46–66)	56 (50–61)	79 (62–86)	74 (70–79)
Soh 2020 [71]	56 (6)	62 (7)	43 (37–46)	45 (38–49)	67 (64–72)	72 (68–78)
Song 2022 [74]	69.9 (7.3)	69.3 (6.8)	—	—	—	—
Susano 2021 [18]	61 (52–67)	65 (60–72)	—	—	—	—
Tobar 2018 [75]	72 (2) ^a^	62 (10)	—	—	—	—
Wood 2017 [19]	63.5 (15) ^a^	69 (22.5)	—	—	—	—
RIGHT						
Ahn 2021 [61]	54.8 (11.9)	54.8 (11.9)	47.8 (11.5)	52.1 (8.7)	72.8 (9.8)	73 (8.9)
Hori 2014 [66]	52 (10.1	55 (9.8)	—	—	—	—
Momeni 2019 [69]	60 (55–67)	63 (56–69)	—	—	—	—
LEFT						
Ahn 2021 [61]	55.7 (11.5)	55.7 (11.5)	48.8 (10.4)	52.1 (9)	73.6 (9.7)	72.5 (8.6)
Hori 2014 [66]	52 (9.5)	54 (9.0)	—	—	—	—
Momeni 2019 [69]	60 (53–66)	62 (55–69)	—	—	—	—

^a^ Estimated from figure; B = baseline rSO_2_; Z is calculated as the measured rSO_2_—threshold rSO_2_ according to the following definition: if M < B, then Z = M; if M ≥ B, then Z = 0. The rSO_2_ desaturation score (DS) for each patient was calculated using the following formula: rSO_2_ DS = (Σ Z) × t, where t = total number of minutes from anaesthesia induction until exit from the operating room. Abbreviations: IQR: interquartile range; rSO_2_: regional cerebral oxygen saturation; SD: standard deviation.

**Table 3 diseases-13-00383-t003:** Methodological quality of included studies according to JBI Critical Appraisal Checklist for Cohort Studies.

Study ID	1	2	3	4	5	6	7	8	9	10	11	Overall
Ahn 2021 [61]										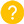		high
Bennett 2021 [62]												low
Chan 2019 [63]												high
Chen 2024 [64]												high
Clemmesen 2018 [60]									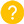	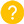		high
Cui 2021 [76]												high
Eertmans 2020 [65]										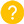		high
Fischer 2022 [72]												moderate
Hori 2014 [66]												moderate
Lim 2020 [67]												high
Mailhot 2019 [68]												high
Momeni 2019 [69]												high
Morimoto 2009 [73]												moderate
Nakano 2021 [70]												high
Schoen 2011 [17]												high
Soh 2016 [77]												high
Soh 2020 [71]												high
Song 2022 [74]												high
Susano 2021 [18]												high
Tobar 2018 [75]												high
Wang 2019 [59]												high
Wood 2017 [19]												high

(1) Were the two groups similar and recruited from the same population? (2) Were the exposures measured similarly to assign people to both exposed and unexposed groups? (3) Was the exposure measured in a valid and reliable way? (4) Were confounding factors identified?5. Were strategies to deal with confounding factors stated? (6) Were the groups/participants free of the outcome at the start of the study (or at the moment of exposure)? (7) Were the outcomes measured in a valid and reliable way? (8) Was the follow up time reported and sufficient to be long enough for outcomes to occur? (9) Was follow up complete, and if not, were the reasons to loss to follow up described and explored? (10) Were strategies to address incomplete follow up utilised? (11) Was appropriate statistical analysis used? 

: yes, 

: no; 
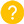
: unclear.

## Data Availability

Data available within the article or its Appendix A.

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
