# Peer review of "Regional Cerebral Oxygen Saturation and Risk of Delirium: A Systematic Review and Meta-Analysis"

_diseases, 2025, doi:10.3390/diseases13120383_

Round 1

Reviewer 1 Report

Comments and Suggestions for Authors

This study reports that both baseline and minimum regional cerebral oxygen saturation (rSO₂) values measured by near-infrared spectroscopy (NIRS) were lower in patients who subsequently developed delirium compared with those who did not. The topic is valuable and clinically relevant; however, several limitations should be addressed to strengthen the manuscript. Specific comments are as follows:

  1. Please expand the discussion on the pathophysiological relationship between cerebral oxygen saturation and delirium to help readers better understand the scientific rationale and background of the study.
  2. Clarify the rationale for excluding randomized trials without control groups.
  3. The discussion is currently rather general. It would be helpful to explicitly state which previous studies your findings are consistent or inconsistent with, and to provide possible explanations for any discrepancies.
  4. The limitations of the study are not clearly described. Please discuss potential sources of bias among the included studies and the possible influence of using different delirium diagnostic tools.
  5. The manuscript notes substantial heterogeneity (I² > 70%); the potential sources of this heterogeneity should be analyzed and discussed in detail.
  6. Some sentences contain grammatical or stylistic issues. It is recommended that the authors seek assistance from a native English speaker or professional language editor to ensure fluency and academic tone.
  7. The subgroup analysis results deserve greater emphasis. For instance, the finding that differences were more pronounced when using the INVOS device and during cardiac surgery should be discussed in depth, along with potential explanations.

Author Response

REVIEWER 1

This study reports that both baseline and minimum regional cerebral oxygen saturation (rSO₂) values measured by near-infrared spectroscopy (NIRS) were lower in patients who subsequently developed delirium compared with those who did not. The topic is valuable and clinically relevant; however, several limitations should be addressed to strengthen the manuscript. Specific comments are as follows:

We would like to thank the reviewer for their valuable comments, which we have taken into account in this revised manuscript. Itemised responses are listed below. All the modifications have been marked in red throughout the manuscript to facilitate review.

  1. Please expand the discussion on the pathophysiological relationship between cerebral oxygen saturation and delirium to help readers better understand the scientific rationale and background of the study.

Author’s answer: Thank you for the suggestion. The authors have added more information in the Discussion section (lines 388 to 404)

  1. Clarify the rationale for excluding randomized trials without control groups.

Author’s answer: Thank you for the opportunity to clarify. The authors have added the following information to the Methods section: “and studies without a pure control group due to intervention bias”

  1. The discussion is currently rather general. It would be helpful to explicitly state which previous studies your findings are consistent or inconsistent with, and to provide possible explanations for any discrepancies.

Author’s answer: Thank you for the suggestion. The authors have revised the discussion, comparing with existing studies and proposing possible explanations for the discrepancies.

The limitations of the study are not clearly described. Please discuss potential sources of bias among the included studies and the possible influence of using different delirium diagnostic tools.

Author’s answer: Thank you for the opportunity to clarify. The authors have added the following information to the discussion section: “The different tools of detecting delirium could also represent a bias in identification. Third, there was a lack of information on some confounding factors, such as comorbidities (e.g. diabetes and hypertension, which can lead to narrowed cerebral arteries through glycosylation and vasculitis), predisposing and precipitating risk factors, and time from NIRS assessment to delirium onset.”

The manuscript notes substantial heterogeneity (I² > 70%); the potential sources of this heterogeneity should be analyzed and discussed in detail.

The subgroup analysis results deserve greater emphasis. For instance, the finding that differences were more pronounced when using the INVOS device and during cardiac surgery should be discussed in depth, along with potential explanations.

Author’s answer: Thank you for the opportunity to clarify. The authors have added the following information to the discussion section: “The forest plots reveal considerable heterogeneity among the included studies, as indicated by I² values of 99% and 98% and highly significant Chi² tests (p < 0.00001). This suggests that the variability in effect sizes is not due to chance but reflects true differences across studies. Potential sources of heterogeneity include differences in patient populations (age, comorbidities, baseline cognitive status), variability in monitoring devices (INVOS vs FORE-SIGHT) with distinct calibration and sensitivity profiles, and differences in clinical contexts such as surgical type and perioperative management. Additionally, inconsistencies in baseline measurement timing and protocols may have contributed to the wide confidence intervals observed. Although subgroup analysis by device type partially explains this variability (p = 0.05), residual heterogeneity remains substantial, indicating that unmeasured confounders or methodological differences persist”

Some sentences contain grammatical or stylistic issues. It is recommended that the authors seek assistance from a native English speaker or professional language editor to ensure fluency and academic tone.

Author’s answer: Thank you for the opportunity to clarify. The article was translated by a specialised medical translator. The authors have attached the revision certificate.

Reviewer 2 Report

Comments and Suggestions for Authors

Dear authors,

this paper addresses a clinically relevant and methodologically interesting question, and the study appears overall well conducted. However, several methodological inconsistencies and descriptive redundancies should be addressed prior to publication.

The Introduction provides a comprehensive background on the use of near-infrared spectroscopy (NIRS) in different surgical settings, together with an overview of delirium epidemiology and risk factors. However, the knowledge gap justifying the present study remains implicit rather than explicitly stated, and the novelty compared with previous reviews is not clearly delineated. I recommend clearly articulating the research gap and specifying how this work advances current evidence.

In the Methods, the PECO framework is only partially accurate: the control group should correspond to patients without delirium, rather than "no comparison". I also suggest including additional details on data extraction procedures (software used, management of missing data, number of reviewers involved).

The Results section is clearly organized and coherent with the reported findings. The authors effectively relate their results to existing literature and provide an interesting interpretation regarding right-hemisphere vulnerability to delirium. Nevertheless, a dedicated section on clinical implications should be added (potential use of perioperative rSO₂ monitoring to identify high-risk patients). Furthermore, the authors should discuss whether the observed mean difference of −2.9% is clinically meaningful in real-world settings.

In the Conclusion, it would be valuable to include a final sentence summarizing the practical or clinical recommendations emerging from the findings.

Author Response

We would like to thank the reviewer for their valuable comments, which we have taken into account in this revised manuscript. Itemised responses are listed below. All the modifications have been marked in red throughout the manuscript to facilitate review.

The authors have reviewed the tables and improved the figures to 600 dpi.

Dear authors,

this paper addresses a clinically relevant and methodologically interesting question, and the study appears overall well conducted. However, several methodological inconsistencies and descriptive redundancies should be addressed prior to publication.

The Introduction provides a comprehensive background on the use of near-infrared spectroscopy (NIRS) in different surgical settings, together with an overview of delirium epidemiology and risk factors. However, the knowledge gap justifying the present study remains implicit rather than explicitly stated, and the novelty compared with previous reviews is not clearly delineated. I recommend clearly articulating the research gap and specifying how this work advances current evidence.

Author’s answer: Thank you for the opportunity to clarify. The authors have added the following in the Introduction section: “To our knowledge, only one meta-analysis has quantified the difference in preoperative cerebral SO2 values between people who did and did not develop postoperative delirium. It included six studies and found lower values in the delirium group [85]. We aimed to update the pooled evidence on this question because several relevant studies have been published in recent years. Specifically, we aimed to evaluate the relationship between cerebral SO2 values obtained by NIRS and the subsequent development of delirium”

In the Methods, the PECO framework is only partially accurate: the control group should correspond to patients without delirium, rather than "no comparison". I also suggest including additional details on data extraction procedures (software used, management of missing data, number of reviewers involved).

Author’s answer: Thank you for the opportunity to clarify. The authors have added more information in the Methods section according to the reviewer’s suggestion.  “To ensure rigour and transparency during data extraction, we employed digital tools that facilitated organisation and reproducibility. We used a customised spreadsheet to systematically record study characteristics, numerical values and statistical estimates, minimising potential errors during synthesis. In parallel, we used the reference management software Zotero to organise bibliographic sources.”

The Results section is clearly organized and coherent with the reported findings. The authors effectively relate their results to existing literature and provide an interesting interpretation regarding right-hemisphere vulnerability to delirium. Nevertheless, a dedicated section on clinical implications should be added (potential use of perioperative rSO₂ monitoring to identify high-risk patients). Furthermore, the authors should discuss whether the observed mean difference of −2.9% is clinically meaningful in real-world settings.

Author’s answer: Thank you for the opportunity to clarify. The authors have added the following in the Discussion section: “Clinically, a difference of nearly 3% in NIRS values may seem modest, but in the context of cerebral autoregulation, this variation may reflect a critical decrease in cerebral perfusion [92]. Because the brain has high metabolic demand and low tolerance to hypoxia, even small sustained reductions in oxygenation could trigger neuronal dysfunction and contribute to the development of delirium [94]. Furthermore, as ours is the first meta-analysis to report this difference, it serves as a starting point for quantifying significantly lower values in patients who develop delirium, encouraging further prospective studies to report more complete data in a larger sample.”

In the Conclusion, it would be valuable to include a final sentence summarizing the practical or clinical recommendations emerging from the findings.

Author’s answer: Thank you for the opportunity to clarify. The authors have added the following in the Conclusion section: “Given that rSO₂ is an objective and easily obtainable measurement, its routine use in hospital settings could improve early detection of delirium risk and optimise clinical decision-making. Studies in other community institutional settings are also warranted.”

Reviewer 3 Report

Comments and Suggestions for Authors

Reviewer’s comments:

1) The authors state that the difference in baseline rSO₂ between the delirium and non-delirium groups is on average 2.92%, which is statistically significant; however, this difference, although confirmed by meta-analysis, may be clinically useless as it is within the known variability between NIRS devices, particularly between INVOS and FORESIGHT. This shortcoming concerns the Results section 3 and the analysis by monitor type, where the authors do not discuss whether this difference is sufficient for clinical use, given that the absolute rSO₂ values between the different devices are not interchangeable. The authors should supplement their discussion in Section 4 with an analysis of whether this 2-3% difference can be considered a reliable predictive threshold if the devices themselves have a systematic difference of 5-10%.

2) Also in section 2, the authors formulate the search query as "NIRS OR Near-infrared spectroscopy OR cerebral oximetr AND delirium OR confusion". However, using the general term “confusion” without specifying the context (e.g., “postoperative confusion” or “ICU confusion”) could have resulted in the inclusion of articles where “confusion” was not diagnosed according to validated criteria for delirium. This potentially undermined the accuracy of the selection. The authors should reformulate the search strategy in the methods or at least discuss this risk in the limitations section.

3) In section 3, the authors note that the difference in rSO₂ is greater when measured with the right sensor, which they attribute to right-hemisphere dominance in attention. However, they do not consider that rSO₂ asymmetry can be a consequence of technical or anatomical factors, such as local edema, atherosclerosis of the internal carotid artery, or even different sensor pressure on the skin. This weakness concerns Table 2 and its caption, which provide separate values for the right and left hemispheres, but do not provide data on symmetry checking or correction for individual anatomical features. The authors should amend the "Limitations" section to include a discussion of the possible influence of local hemodynamic or technical artifacts on the interpretation of rSO₂ asymmetry.

4) Finally, in Section 3, the analysis of rSO₂ trough values is based on only 8 studies, and the authors acknowledge high heterogeneity (I² = 83-97%). However, they do not explain whether trough values were defined as the absolute minimum over the entire observation period or only at specific time points. This is critical because interval measurements (e.g., every 5 minutes) may miss the true trough. This shortcoming applies to Table 2 and the description of measurement methods. The authors should clarify the criteria for determining "minimum rSO₂" in the "Materials and Methods" section and, if possible, conduct a sensitivity analysis that separates data by type of monitoring (continuous vs. interval).

Author Response

We would like to thank the reviewer for their valuable comments, which we have taken into account in this revised manuscript. Itemised responses are listed below. All the modifications have been marked in red throughout the manuscript to facilitate review.

Reviewer’s comments:

The authors state that the difference in baseline rSO₂ between the delirium and non-delirium groups is on average 2.92%, which is statistically significant; however, this difference, although confirmed by meta-analysis, may be clinically useless as it is within the known variability between NIRS devices, particularly between INVOS and FORESIGHT. This shortcoming concerns the Results section 3 and the analysis by monitor type, where the authors do not discuss whether this difference is sufficient for clinical use, given that the absolute rSO₂ values between the different devices are not interchangeable. The authors should supplement their discussion in Section 4 with an analysis of whether this 2-3% difference can be considered a reliable predictive threshold if the devices themselves have a systematic difference of 5-10%.

Author’s answer: Thank you for the opportunity to clarify. The authors have added the following in the Discussion section: “Clinically, a difference of nearly 3% in NIRS values may seem modest, but in the context of cerebral autoregulation, this variation may reflect a critical decrease in cerebral perfusion [92]. Because the brain has high metabolic demand and low tolerance to hypoxia, even small sustained reductions in oxygenation could trigger neuronal dysfunction and contribute to the development of delirium [94]. Furthermore, as ours is the first meta-analysis to report this difference, it serves as a starting point for quantifying significantly lower values in patients who develop delirium, encouraging further prospective studies to report more complete data in a larger sample.”

2) Also in section 2, the authors formulate the search query as "NIRS OR Near-infrared spectroscopy OR cerebral oximetr AND delirium OR confusion". However, using the general term “confusion” without specifying the context (e.g., “postoperative confusion” or “ICU confusion”) could have resulted in the inclusion of articles where “confusion” was not diagnosed according to validated criteria for delirium. This potentially undermined the accuracy of the selection. The authors should reformulate the search strategy in the methods or at least discuss this risk in the limitations section.

Author’s answer: Thank you for the opportunity to clarify. The authors included the term “confusion” to extend the number of studies, but to avoid the bias detected by the reviewer, only those studies in which delirium was assessed using validated diagnostic instruments were included, as indicated in the inclusion criteria. This is also reflected in the results table. In any case, the authors have added the following in the limitation section: “The different tools of detecting delirium could also represent a bias in identification”

3) In section 3, the authors note that the difference in rSO₂ is greater when measured with the right sensor, which they attribute to right-hemisphere dominance in attention. However, they do not consider that rSO₂ asymmetry can be a consequence of technical or anatomical factors, such as local edema, atherosclerosis of the internal carotid artery, or even different sensor pressure on the skin. This weakness concerns Table 2 and its caption, which provide separate values for the right and left hemispheres, but do not provide data on symmetry checking or correction for individual anatomical features. The authors should amend the "Limitations" section to include a discussion of the possible influence of local hemodynamic or technical artifacts on the interpretation of rSO₂ asymmetry.

Author’s answer: Thank you for the suggestion. The authors have added the following in the limitations section: “It was not possible to perform a systematic check for rSO₂ symmetry or apply corrections for individual anatomical factors due to the design and scope of the study. Likewise, we could not rule out the potential influence of local hemodynamic or technical artifacts (e.g. oedema, internal carotid artery atherosclerosis or variations in sensor pressure on the skin), which may have affected the interpretation of the observed asymmetry”

4) Finally, in Section 3, the analysis of rSO₂ trough values is based on only 8 studies, and the authors acknowledge high heterogeneity (I² = 83-97%). However, they do not explain whether trough values were defined as the absolute minimum over the entire observation period or only at specific time points. This is critical because interval measurements (e.g., every 5 minutes) may miss the true trough. This shortcoming applies to Table 2 and the description of measurement methods. The authors should clarify the criteria for determining "minimum rSO₂" in the "Materials and Methods" section and, if possible, conduct a sensitivity analysis that separates data by type of monitoring (continuous vs. interval).

Author’s answer: Thank you for the suggestion. The authors have added the following in the Methods section:

  • “Baseline NIRS: before induction of anaesthesia with the patient breathing ambient air and 1 minute after placement of the measurement sensor
  • Minimum NIRS: the lowest recorded value during assessment
  • Maximum NIRS: the highest recorded value during assessment”

In addition, the authors have performed the same analyses according to measurement type (continuous and interval), adding it in the Results and Discussion section, as well as the forest plots to the Supplementary File (Figures S6-S9): “Finally, we conducted the same analyses according to the type of measurement (continuous or intermittent). There was a higher MD in rSO2 values with continuous assessment (MD −4.15, 95% CI −6.95 to −1.35; Tau² = 3.37; Chi² = 6.37, df = 4, P < 0.00001; I² = 37%; 5 studies, 647 participants) compared to intermittent assessment (MD −2.54, 95% CI −4.51 to −0.57; Tau² = 6.11; Chi² = 36.36, df = 8, P < 0.00001; I² = 78%; 9 studies, 2437 participants), also reflecting the heterogeneity of the types of observation measures (Figure S6). The results were similar in the remaining analyses for baseline and minimum values (Figures S7-S9).”

Reviewer 4 Report

Comments and Suggestions for Authors

The authors performed a systematic review and meta-analysis to corelate the risk of delirium and regional cerebral oxygen saturation. I have several concerns about the methodology and the impact of confounding factors on the results.

  1. How did the authors differentiate Emergence agitation vs Emergence delirium which are used interchangeably in clinical practice. EA has been excluded from search result from the Key word
  2. In the exclusion criteria, I don't see patient with previous ED or Post traumatic disorder excluded from the analysis since they play an important role for the onset of ED.
  3. The authors mentioned the introduction about the use of biomarkers, none of this has been described in the discussion part.
  4. The authors didn't describe how cerebral hypoxia triggers delirium. The role of reactive oxygen species and other underlying pathophysiology needs to be discussed.
  5. Emergence delirium is most common in older population with significant comorbidities, especially diabetes and Hypertension who have narrowed cerebral arteries due to glycosylation and vasculitis. This would lead to potential bias of the results.

Author Response

The authors performed a systematic review and meta-analysis to corelate the risk of delirium and regional cerebral oxygen saturation. I have several concerns about the methodology and the impact of confounding factors on the results.

We would like to thank the reviewer for their valuable comments, which we have taken into account in this revised manuscript. Itemised responses are listed below. All the modifications have been marked in red throughout the manuscript to facilitate review.

The authors have reviewed the tables and improved the figures to 600 dpi.

  1. How did the authors differentiate Emergence agitation vs Emergence delirium which are used interchangeably in clinical practice. EA has been excluded from search result from the Key word

Author’s answer: Thank you for the suggestion. The authors did not include the concept of EA, as we considered ED to be more comprehensive by encompassing both hyperactive delirium—typically referenced by EA—and hypoactive delirium. However, in accordance with the reviewer’s suggestion, we have added this item to the Limitations section. “The first limitation of this review is that our search was restricted to the predefined terms and the selected databases, which means we may have missed some relevant studies published in other sources or using different terminology.”

  1. In the exclusion criteria, I don't see patient with previous ED or Post traumatic disorder excluded from the analysis since they play an important role for the onset of ED.

Author’s answer: Thank you for the opportunity to clarify. The authors did not include participants with previous delirium, as this could bias the baseline results.

  1. The authors mentioned the introduction about the use of biomarkers, none of this has been described in the discussion part.

Author’s answer: Thank you for the opportunity to clarify. The authors have removed the following sentence from the discussion section to focus the manuscript strictly on the use of NIRS.. “Biomarkers such as serum, salivary or imaging markers play a key role in identifying delirium and evaluating its risk”.

  1. The authors didn't describe how cerebral hypoxia triggers delirium. The role of reactive oxygen species and other underlying pathophysiology needs to be discussed.

Author’s answer: Thank you for the suggestion. The authors have added more information in the Discussion section (lines 388-404) 

  1. Emergence delirium is most common in older population with significant comorbidities, especially diabetes and Hypertension who have narrowed cerebral arteries due to glycosylation and vasculitis. This would lead to potential bias of the results.

Author’s answer: Thank you for the opportunity to clarify. The authors have added the following in the limitations section: “Third, there was a lack of information on some confounding factors, such as comorbidities (e.g. diabetes and hypertension, which can lead to narrowed cerebral arteries through glycosylation and vasculitis), predisposing and precipitating risk factors, and time from NIRS assessment to delirium onset.”

Round 2

Reviewer 1 Report

Comments and Suggestions for Authors

Thank you to the author for the thorough revisions. All the issues have been fully addressed, and I have no further comments. In my view, the manuscript demonstrates a high level of innovation.  This manuscript fully meets the standards for publication in the journal. I believe it is suitable for acceptance. It has been a pleasure to engage in academic exchange with the author through this manuscript. Thank you for your respectful response to my suggestions.

Author Response

Thank you to the author for the thorough revisions. All the issues have been fully addressed, and I have no further comments. In my view, the manuscript demonstrates a high level of innovation.  This manuscript fully meets the standards for publication in the journal. I believe it is suitable for acceptance. It has been a pleasure to engage in academic exchange with the author through this manuscript. Thank you for your respectful response to my suggestions

Author’s answer: Thank you

Reviewer 4 Report

Comments and Suggestions for Authors

The authors just added my major concerns to the limitation section and try to ignore answering the relevant questions and adding few more details to improve the article quality

Author Response

The authors just added my major concerns to the limitation section and try to ignore answering the relevant questions and adding few more details to improve the article quality.

Author’s answer: Thank you, the authors feel that we have replied to all comments and suggestions. We have rechecked the comments as follows:

Comment 1: How did the authors differentiate Emergence agitation vs Emergence delirium which are used interchangeably in clinical practice. EA has been excluded from search result from the Key word

Author’s answer R1: Thank you for the suggestion. The authors did not include the concept of EA, as we considered ED to be more comprehensive by encompassing both hyperactive delirium—typically referenced by EA—and hypoactive delirium. However, in accordance with the reviewer’s suggestion, we have added this item to the Limitations section. “The first limitation of this review is that our search was restricted to the predefined terms and the selected databases, which means we may have missed some relevant studies published in other sources or using different terminology.”

Author’s answer R2: According to the PROSPERO registry, authors should not modify the search equation, we stand by the answer given in R1.

Comment 2: In the exclusion criteria, I don't see patient with previous ED or Post traumatic disorder excluded from the analysis since they play an important role for the onset of ED.

Author’s answer R1: Thank you for the opportunity to clarify. The authors did not include participants with previous delirium, as this could bias the baseline results, we have included it as a exclusion criteria.

Author’s answer R2: Thank you for the opportunity to clarify, we have included it as an exclusion criteria.

Comment 3: The authors mentioned the introduction about the use of biomarkers, none of this has been described in the discussion part.

Author’s answer: Thank you for the opportunity to clarify. The authors have removed the following sentence from the discussion section to focus the manuscript strictly on the use of NIRS “Biomarkers such as serum, salivary or imaging markers play a key role in identifying delirium and evaluating its risk”.

Author’s answer R2: We stand by the answer given in R1.

Comment 4: The authors didn't describe how cerebral hypoxia triggers delirium. The role of reactive oxygen species and other underlying pathophysiology needs to be discussed.

Author’s answer: Thank you for the suggestion. The authors have added more information in the Discussion section

Author’s answer R2: We stand by the answer given in R1.

Comment 5: Emergence delirium is most common in older population with significant comorbidities, especially diabetes and Hypertension who have narrowed cerebral arteries due to glycosylation and vasculitis. This would lead to potential bias of the results.

Author’s answer R1: Thank you for the opportunity to clarify. The authors have added the following in the limitations section: “Third, there was a lack of information on some confounding factors, such as comorbidities (e.g. diabetes and hypertension, which can lead to narrowed cerebral arteries through glycosylation and vasculitis), predisposing and precipitating risk factors, and time from NIRS assessment to delirium onset.”

Author’s answer R2: We stand by the answer given in R1. Most studies do not report this data, and it is reflected as one of the points to be analyzed in future studies.